# Energy Potential Assessment of Excavated Landfill Material: A Case Study of the Perm Region, Russia

**Iuliia Shcherbinina [1], Stepan Polygalov [1], Galina Ilinykh [1], Vladimir Korotaev [1], Natalia Sliusar [1,*], Ivana Mihajlovic [2] and Nemanja Stanisavljevic [2]**

[1] Environmental Protection Department, Perm National Research Polytechnic University, 614990 Perm, Russia; julsherbinina@gmail.com (I.S.); polyste17@mail.ru (S.P.); galina.perm.59@yandex.ru (G.I.); korotaev@pstu.ru (V.K.)

[2] Department of Environmental Engineering, Faculty of Technical Sciences, University of Novi Sad, 21000 Novi Sad, Serbia; ivanamihajlovic@uns.ac.rs (I.M.); nemanjastanisavljevic@uns.ac.rs (N.S.)

* Correspondence: nnsliusar@pstu.ru; Tel.: +79-223-202-560

**Abstract:** The paper presents results of field and laboratory studies of thermal characteristics to excavated landfill waste in Perm region, Russia. The peculiarity of the study includes the following aspects: waste composition with a high share of polymers, the climatic conditions of the territory and the lack of engineering infrastructure at the waste disposal facility. When determining the waste composition and thermal properties of waste, it is proposed to include a stage of removal of contamination from landfilled waste fraction, since their share of contamination can reach up to 33%. This stage will allow researchers to adjust the net calorific value of the excavated waste without overestimation, which may affect decision-making when implementing waste management technology. Among combustible components with the highest moisture content are waste paper (69.1%) and diapers (65.8%), whereas wood (11.2%), PET bottles (3.1%) and other 3D plastics (13.4%) have rather low ash content on a dry basis. Calculation of thermal properties and analysis of the energy potential of the waste samples was conducted based on the obtained data. The calorific value of the individual components and excavated waste depends not only on the moisture and ash content of the individual components, but also on the presence of contaminants. The average net calorific value of the excavated waste is 4.9 MJ/kg, and for the separate mixture of combustible components, it is 7.5 MJ/kg at a moisture content of 44%. Excavated landfill waste can be regarded as a resource for the manufacture of secondary fuel only after pretreatment that includes at least sorting and drying. The results of this study may be useful in developing technologies needed to eliminate old MSW dumps and old landfills, for the development of the concept of circular economy and prevention of environmental degradation problems.

**Keywords:** municipal solid waste; landfill mining; calorific value; moisture content; ash content

## 1. Introduction

In recent years, there has been a steady worldwide increase in the volume of waste being generated, including municipal solid waste (MSW). In the MSW industry, several waste management methods are used, such as composting, thermal treatment, landfilling, and extraction of secondary raw materials for the subsequent manufacture of new products. Nevertheless, disposal of MSW in landfills and dumps remains the basic model of waste management for many countries, including Russia. This method is the most common and cost efficient, taking into account the current development level of the disposal technologies in use. However, leachate and biogas formed during the waste decomposition pollute the hydrosphere and atmosphere. This not only contributes to environmental degradation, but also poses a threat to public health. In addition, one of the drawbacks of landfills is the loss of access to significant amounts of land which can add up to thousands of hectares

of land nationwide for an indefinite period. It is also necessary to understand that in the Russian Federation, for example, aside from landfills that are authorized and at least minimally equipped with facilities, there are a large number of illegal dumps, which have a significantly greater impact on the environment.

In addition, landfills, which were built decades ago, are often located in areas that absolutely cannot be used for waste disposal according to modern regulations. In particular, it is forbidden to place landfills and dumps near residential buildings, in water protection zones, etc. In this regard, along with the traditional reclamation of landfills and dumps on-site, some facilities are recommended for elimination, which includes excavation and redisposal of the entire waste stockpile, and restoration of the territory. The resulting excavated landfill material is usually redisposed at another facility, which imposes huge costs and does not solve the waste problem.

A promising alternative is processing landfill materials in order to extract potentially useful components from it and further investigate them. Excavation of waste disposal facilities with landfill material treatment will not only free land plots from waste deposits, but also make it possible to obtain secondary material and/or energy resources from the excavated soils.

Excavation of old dumps and landfills has been addressed in the research of scientists from numerous countries [1–8]. The motivation for many projects [9–14] was land reclamation and end of landfills' operational life. Most of the work is based on results of experimental studies of the composition and properties of excavated landfill materials and the assessment of its resource potential. Part of the projects [15,16] presents the results of studies focused on physicochemical processes that occur during biodegradation of waste in the landfill body.

One of the most suitable way for landfill mining projects valorization is energy recovery from excavated waste [14,17,18]. An assessment of the resource potential of excavated landfill materials is presented in a study by Finnish scientists [19]. The energy potential of waste taken from the landfill is compared with the energy level of fuel acquired from waste that has not been disposed of. Ultimately, the elimination of MSW landfills offers new commercial opportunities related to reclamation. The land can be used again for housing, industrial property development or other forms of construction.

The analysis of bulk density, moisture values, and component composition of the fine fraction of less than 10 mm removed from the MSW landfill is given in study [20]. The results of this study are of paramount importance for assessing the energy potential and the ability to extract material.

One of the promising areas of beneficial use for excavated waste is energy recovery of its most high-calorie components. For example, in the results of studies [21] the component composition of solid fuel from excavated waste was determined as follows: films—29.4%, PET bottles—1.9%, other plastics and synthetic textiles—35.2%, leather—7.5%, wood—13.0%, paper—13.0%. The calorific value of solid fuel from excavated waste was determined experimentally and is equal to 21.5 and 23.6 MJ/kg [22]. Furthermore, thermal property studies [23,24] of "fresh" MSW made it possible to develop an algorithm for assessing the energy potential of heterogeneous flows, which can include landfill soil.

However, the results of foreign studies of the composition and properties of landfill materials can only be extrapolated for Russian conditions in limited ways. Excavated landfill material has an inconsistent and heterogeneous composition, which depends on the timeframe and technology of disposal, the presence or absence of various engineering structures at the landfill, the types and composition of landfill waste, the climate of the territory, and other factors. The existing waste management system and its evolution in earlier years also play a significant role. In particular, since 2005, the burial of untreated waste has been prohibited in the European Union; accordingly, the composition of landfill materials "younger" than this date at European facilities will differ significantly from the composition of disposal waste with the same age at Russian landfills. Given the exponential increase in the consumption of polymer materials in Russia in the 2000s and

early 2010s, as well as the low development level of separate waste collection and disposal technologies, the composition and properties of Russian landfill materials aged 5–10 years from the moment of disposal will differ significantly from previously published data for European countries. Studies on waste excavation in Russia have appeared only relatively recently [25,26]. Studies [7,26] present a qualitative characteristic of old landfill stockpiles. It reveals that at the date of MSW disposal, a fine fraction begins to form (the so-called landfill subsoil materials), which proportion increases with the sampling depth and the age of the waste. This means that as it ages beyond 25 years, landfill material acquires an average of 85–90% fine fraction.

The aim of this paper is to study of the energy potential of excavated landfill material. Research included sampling, analysis of sample component composition and thermal properties of individual components, as well as assessment of the energy potential of excavated landfill materials. Among the specific objectives of the study is to analyze the thermal properties of excavated waste with and without presence of adhering contaminants. Specific excavations of excavated material are of particular importance for the development of the concept of improved landfill mining because the exploitation of excavated material must be individually adapted to the material characteristics. This way of excavating old landfills in combination with the use of energy from excavated material is important for the development of the concept of circular economy and prevention of environmental degradation problems.

## 2. Materials and Methods

### 2.1. Sampling

Waste sampling was conducted in December 2018 at several points (Figure 1 and Table 1) of MSW disposal site (operational 1992–2023) located in the Perm region, Russia.

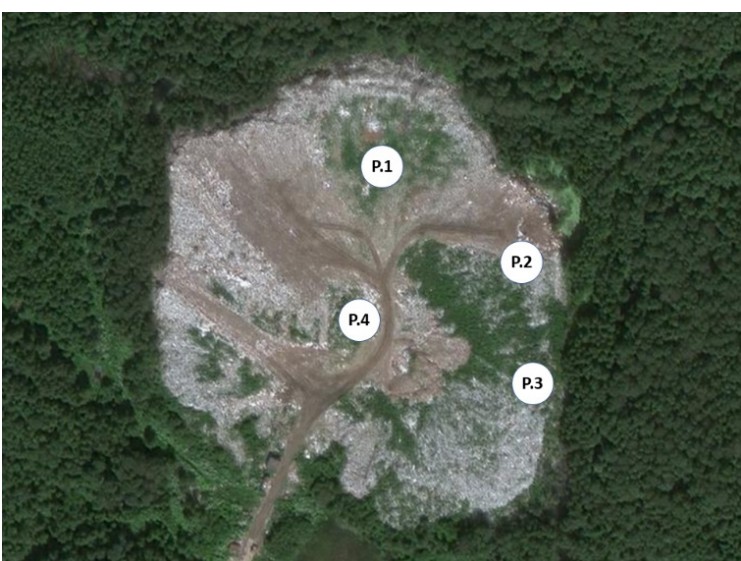

**Figure 1.** Location of sampling points at the MSW disposal facility.

**Table 1.** Sampling program.

| Sampling Depth, m | Point 1 | Point 2 | Point 3 | Point 4 |
|---|---|---|---|---|
| 1.0–1.5 | Sample 1.1 | | Sample 3.1 | Sample 4.1 |
| 2.0–2.5 | | Sample 2.1 | Sample 3.2 | |
| 2.5–3.0 | Sample 1.2 | Sample 2.2 | | Sample 4.2 |
| 3.5–4.0 | | | Sample 3.3 | |
| 4.0–4.5 | | | | Sample 4.3 |
| 4.5–5.0 | | Sample 2.3 | Sample 3.4 | |

The climate of Perm region, Russia is characterized by snowy, cold, and long winters and moderately warm summers. Terrain features have a significant impact on climatic conditions due to the barrier influence of the Ural Mountains. In consequence of this influence the climate of the studied region is characterized by significant amount of precipitation (650 mm) and low average annual temperatures ($0 \pm 2$ °C).

As with many other waste disposal sites in the Perm region and Russia, there is no engineering infrastructure at the studied waste disposal facility (baseliner, intermediate covering, final cover, leachate, and landfill biogas collection systems, etc.).

Waste sampling at the landfill was carried out by a bucket excavator along the profile of the waste stockpile with an interval of 0.5–1.0 m vertically. Algorithm of laboratory samples preparation is shown in Figure 2. At each point and precise depth, 2–4 samples were taken, weighing 10–30 kg each. Total of 24 samples with a final weight of about 450 kg were transported to the laboratory for further analysis.

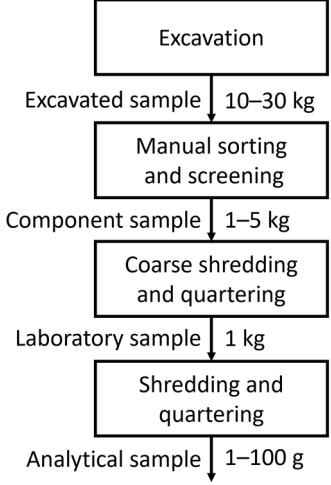

**Figure 2.** Algorithm of laboratory samples preparation.

### 2.2. Analysis of Sample Component Composition

Each sample was manually sorted into the following components: waste paper, polyethylene (PE) and polypropylene (PP) films, metallized and multilayer films, PET bottles, other 3D plastics, textiles, wood, glass, metals, inert materials, fine fraction less than 20 mm, and unidentifiable materials [27].

Laboratory samples were prepared according to GOST 33509-2015 (EN 15443:2011). The determination of each MSW component's composition was carried out in its natural wet state. After separation of samples into individual components, samples with a mass of one kilogram were taken of individual components to measure their thermal properties.

### 2.3. Analysis of Thermal Properties of Individual Components

To determine the thermal properties of the components, laboratory samples were prepared according to GOST 33510-2015 (EN 15413:2011). Analysis of individual components thermal properties included the determination of: moisture content, degree of contamination and ash content.

The moisture content of the waste was determined on the basis of GOST 33512.3-2015 (EN 15414-3:2011) "Solid fuel from household waste. Determination of moisture content by drying. Part 3. Moisture analysis" [28]. Samples were dried in an oven at a temperature of $105 \pm 2$ °C until the sample mass became unchanged. The moisture content was calculated, based on the mass of the sample before and after drying.

The dried components were weighed and then manually cleaned of adhering dirt. Contaminants, that stuck to waste components surface or were inside bags, cans, bottles, and containers, were removed with a brush. The components themselves and the separated contaminants were weighed individually. Based on the data obtained, the percentage of

contamination of the components was calculated. The study to determine the contamination of waste components were carried out to show the differences in waste properties without and with contamination.

Ash content was calculated based on GOST 33511-2015 (EN 15403:2011) "Solid fuel from household waste. Ash determination" [29]. The components were weighed before and after ashing in the muffle furnace at a temperature of 550 °C. The ash content of the components on a dry basis was calculated. The ash content was determined only for samples of combustible materials. Calorific value of absolutely non-combustible components, such as metal, glass, and inert materials after drying and separating contaminants was taken to be equal to zero.

### 2.4. Assessment of the Energy Potential of Excavated Landfill Materials

Calculation of the calorific value of all waste the following equation was used [30]:

$$Q_i^r = \sum [K_i^r (1 - W_i^r / 100)\left(1 - A_i^d / 100\right) \times Q_i^{daf}] - 0.02442 \times \sum (K_i^r \times W_i^r) \quad (1)$$

where: $K_i^\gamma$—mass fraction of the $i$ component, %; $W_i^\gamma$—moisture content of the $i$ component; $A_i^d$—ash content on a dry basis of the $i$ component; $Q_i^{daf}$—the net calorific value for dry ash-free basis of the $i$ component, MJ/kg.

Separated contaminants after drying were also considered as a separate component. In order to assess the correlations between the experimental data, statistical analyses were performed using IBM SPSS (software package version 25, IBM North America, New York, NY, USA). Pearson correlation matrix was obtained considering selected variables (component composition, ash content, moisture content, calorific value and calorific value without contamination). A confidence level of 95% was selected for statistical assessment.

## 3. Results and Discussion

To assess the energy potential of excavated waste samples from the MSW landfill, an analysis of thermal properties was undertaken to calculate component composition, moisture content, ash content and contamination of the components. Changes in the waste property on the depth in the landfill body was not studied since the waste disposal on the studied MSW disposal site was chaotic and unsystematic by depth and location

### 3.1. Component Composition of Excavated Waste

Based on the studies performed, the component composition of individual samples of excavated waste was determined, as was the composition averaged over all samples (Figure 3).

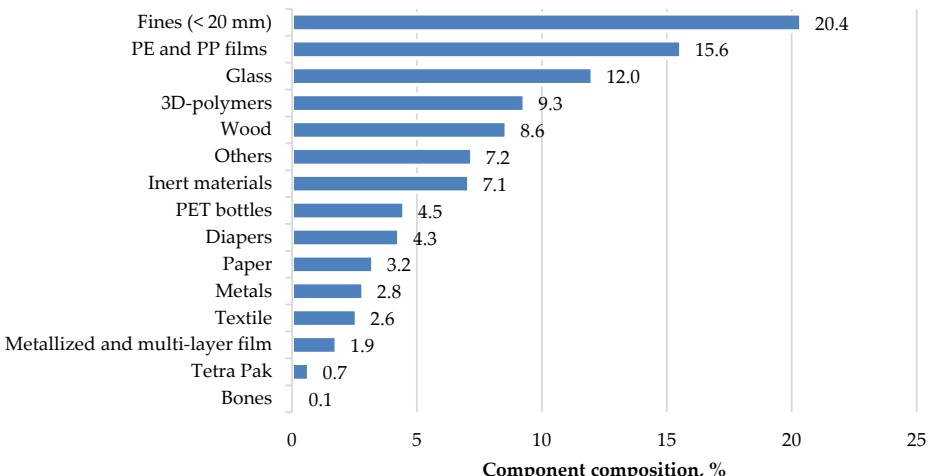

**Figure 3.** Average component composition of excavated waste.

The highest content value in the composition of excavated waste was determined for landfill soil (fines less than 20 mm)—17.7–27.7%; glass—11.4–13.5%; and polymers—26.3–40.2%. The absence of separate waste collection on the territory was resulted to a high share of packaging (PE and PP films, 3D polymers, PET bottles, metallized and multilayer film), so the excavated landfilled waste can be considered as an energy resource. The proportion of combustible components in the excavated waste for various samples was 25–59% of the total MSW composition, with an average of 47%. The age of the waste (4–9 years old) was confirmed by visual analysis of newspaper clippings and food bags. In this regard, it can be concluded that components of biogenic origin (wood, textiles, books and magazines with dense folded sheets) did not decompose completely during this time, whereas food waste as well as loose paper and cardboard did, so the proportion of landfill soil is lower compared to the results of earlier studies [8,21,22,31,32]. Accordingly, these research results confirm that as the waste ages, the proportion of landfill soil increases.

Table 2 summarizes the component composition of the excavated waste, and the characteristics of MSW disposal facilities in different countries, including Austria, Hungary, India, and China.

**Table 2.** Summary data on the component composition of excavated waste and characteristics of MSW disposal facilities in different countries.

| Name of the Object (Country) | Landfill Lower Austria, Austria [8] | Landfill Styria, Austria [8] | Landfill Kudjape, Estonia [21] | Landfill Delhi, India [31] | Landfill Hyderabad, India [31] | Landfill Kadapa, India [31] | Landfill Debrecen, Hungary [22] | Landfill Inchuan, China [32] |
|---|---|---|---|---|---|---|---|---|
| Object characteristics | | | | | | | | |
| Years of exploitation | 1982–2003 | 1979–2004 | 1970–2009 | since 1994 | since 1999 | since 1965 | since 1993 | 1989–2004 |
| Object area, ha | 7.8 | n/a | 3.9 | 16.16 | 123.22 | 4.04 | 20.1 | 11.3 |
| Sampling depth, m | 7–18 | 5–6 | 5 | 4–5 | 4–5 | 4–5 | up to 12 | up to 24 |
| Component composition, % | | | | | | | | |
| Waste paper | 1.1 | 1.2 | 3.4 | n/a | n/a | n/a | 4.9 | 0.2 |
| Plastics | 10.3 | 8.2 | 17.4 | 3.3 | 2.7 | 3.7 | 20.6 | 10.6 |
| Textile | 4.4 | 2.7 | 17.4 | 0.8 | 1.4 | 1 | 4.6 | 1.5 |
| Wood | 5.4 | 1.6 | 3.4 | 0.2 | 1.5 | 1.3 | 5.3 | 2.4 |
| Leather | 5.4 | 1.6 | n/a | n/a | n/a | n/a | 5.3 | n/a |
| Rubber | 5.4 | 1.6 | 2.0 | n/a | n/a | n/a | 5.3 | n/a |
| Inert materials | 1.1 | 1.8 | 10.1 | 23.4 | 15.4 | 16.2 | 11.2 | 8.3 |
| Glass | 0.1 | 0.1 | 0.4 | 0.2 | 2.6 | 1.7 | n/a | 0.6 |
| Metals | 3.5 | 0.9 | 2.7 | n/a | n/a | n/a | 2.9 | 0.4 |
| Other | 6.1 | 4.0 | 6.6 | 0.2 | 3.3 | 0.9 | 0.5 | 1.0 |
| Fines less than 40 mm | 68.0 | 79.5 | 54.0 | 71.9 [1] | 73.1 [1] | 75.2 [1] | 50.0 [2] | 75.0 [2] |

[1] landfill soil (fraction size not specified). [2] fraction less than 20 mm.

Based on the results of Table 2, it can be concluded that most of the waste is represented by landfill soil (50–80%), although waste sampling was mainly carried out at a depth of 4–6 m, with an waste stockpile height of up to 70 m. Landfill soil can be used for various purposes: as a sprinkling between layers (shelter, backfill) on the site itself, or as compost with the addition of another subsoil (soil, fertilizer of a different origin).

Study object, MSW disposal site, is on operation, so biogenic origin components have not yet had time to decompose completely in the landfill body. MSW is brought to the disposal site without preliminary waste sorting.

However, according to studies [8,21,22,31,32] determining its elemental composition, landfill soil has a high content of organic substances, chlorides, sulfates, etc., so rather serious questions arise that must be taken into account before using it for agricultural purposes. A high proportion of landfill soil in the composition of excavated waste is linked to the age of the waste in the landfill (15–55 years) and the composition of the landfilled waste. As previously mentioned, a prohibition of valuable materials (polymers, paper, metals, glass) at MSW landfills has been in effect in European countries since 2005.

For India, the proportion of combustible fraction does not exceed 6%, which is attributable to the country's lower level of economic development and the early opening of local MSW disposal facilities. Some Indian facilities have been in operation since 1965

when there were no packaging materials in the local waste [33], though the use of plastic has increased since the end of 1980s [21]. In addition, plastics are stable materials at MSW disposal facilities when compared to organic waste, paper, cardboard, and wood, which decompose after some time.

In southern countries, for example, India and China, the share of the small fraction ranges from 71 to 75% [31,32], which is higher compared to European countries. For example, in Hungary, the share of the small fraction is 50%, in Estonia—54%, in Austria—68% [8,21,22]. It depends on the fact that in European countries the share of waste packaging (plastic, glass, paper, metal) is significantly predominant. For example, in Austria the share of packaging is 15.0%, in Hungary—28.4%, in Estonia—23.9% [8,21,22], even considering the ban for valuable components landfilling.

The maximum proportion of scrap metal is observed in the Lower Austria landfill (3.5%). Such low values are due to the fact that metals are selected from the waste stream for disposal earlier on at the separate collection stage.

It is important to note that in most publications [8,22,31,32], plastics are listed in one category, and only in some studies [21] are they sorted into "soft pastics", i.e., films (13–22% of the total coarse fraction greater than 40 mm), and mixed plastics together with synthetic textiles (16–22% of the composition of the MSW fraction larger than 40 mm). A similar observation can be made for metals. In some papers [21,22], various scrap metals were indicated (iron, aluminum, copper, and stainless steel). Compiling a detailed list of the waste components identified allows a more accurate assessment of the material and energy potential of the MSW. This is important since the requirements for quality of secondary raw materials and thermal properties of individual components often differ significantly.

### 3.2. Moisture Content of the Components

The moisture values of the components play an important role in the combustion process, since high moisture content slows down and complicates the burning. Moreover, each component has a different structure, porosity and hygroscopicity, which affects its moisture value. The results of determining the moisture content of individual components are presented in Figure 4.

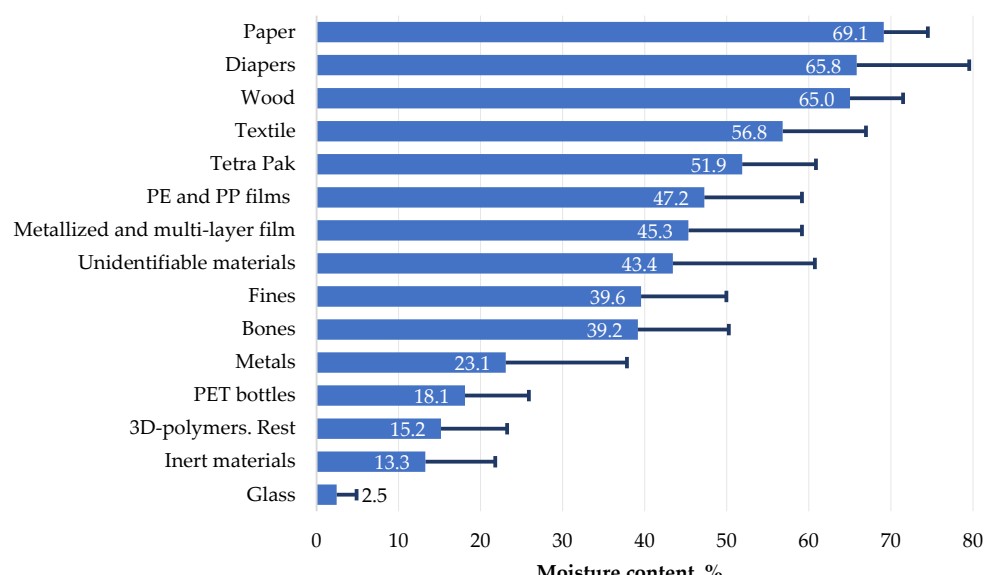

**Figure 4.** Moisture content of individual components (averaged data for all samples with error bars).

The components with the highest moisture value included waste paper (69.1%), diapers (65.8%) and wood (65.0%) (Figure 2). As noted above, studies of the thermal properties of "fresh" waste were performed [23,24], and if the moisture content of individual components of "fresh" waste and excavated waste is compared, the moisture content of some

components, such as diapers, is almost identical. The moisture content of waste paper and wood in the composition of "fresh" waste is respectively 2.2 times and 4 times lower than in excavated waste. This is due to the fact that waste paper, cardboard, and wood are buried in contact with high-moisture components for a long time, which leads to full or practically full moisture saturation of these components. The lowest humidity value (2.5%) is characteristic of glass, for which moisture content is determined by the presence of adhering contaminants. Despite the fact that the polymer films themselves are not hygroscopic, as well as 3D plastics, moisture droplets and wet pieces of landfill material adhere well to the large surface of the films.

The moisture content of individual components by sampling points 1 to 4 is shown in Figure 5. It can be concluded that the moisture indicators of the components are relatively constant, regardless of the sampling location.

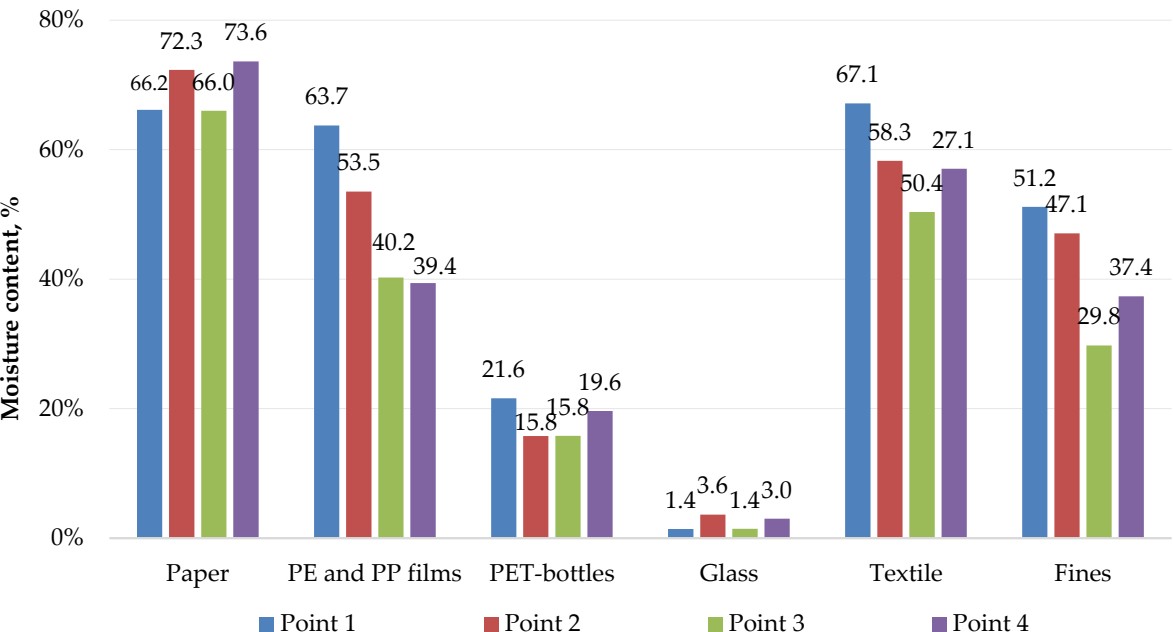

**Figure 5.** The moisture content of the components at individual sampling points.

It should also be noted that moisture content, ash content and component contamination were determined for each component at each sampling point. However, six components with the most clearly differentiable indicators (waste paper, PE and PP films, PET bottles, glass, textiles, and sifting) were selected for comparison.

### 3.3. Component Contamination

Before determining the ash content of the samples, the components were cleaned of contaminants, and the proportion of contaminants was determined. This stage was necessary in order to fully understand the true content of combustible components in the excavated waste, and its energy potential. As previously stated, wet and ash contaminants adhering to PET bottles (a component with a high calorific value) will obviously lead to an underestimation of its thermal performance, which must be taken into account in further calculations.

In order to assess the contribution of contaminants to the inaccuracy of further calculations, moisture content (54.3%) and ash content on a dry basis (78.5%) were also determined for the contaminant itself (most likely a mixture of fine debris, sand, and decaying particles of food with other components of biological origin).

Based on the weight of the components before and after cleaning, the percentage of contamination was calculated in the components' dried state. The results for the individual components are presented in Figure 6.

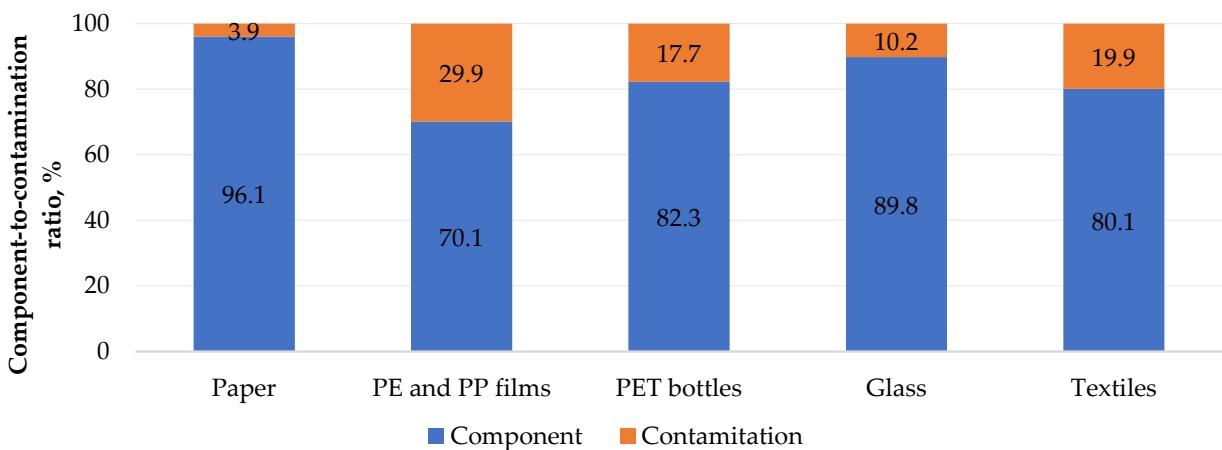

**Figure 6.** Average contamination of components (on a dry basis).

The fraction of adhering contaminants on the components on a dry basis ranges from 3.8% for inert materials, to 33.4% for metallized and combined films. Most often, contaminants form inside packages, bottles, cans, containers, lids, pockets, etc., as well as in the folds of crumpled bottles, packages, etc., which is proved by the results of studies: the average pollution of glass is 10.2%, and for PE and PP films it is 29.9%.

Contaminants reach and adhere to these components during the stage of mixed waste collection, the transportation and compaction in garbage trucks and at MSW disposal sites (i.e., during decomposition of organic waste and compaction of the waste mass). Accordingly, the whole form components (stones, wood, bones, and paper) have the lowest contamination values. Thus, it is evident that size, shape, volume, and structure of components affect their contamination. This must be taken into account both in determining the component composition of the waste, and its thermal properties.

*3.4. Component Ash Content*

The results of ash content studies of the previously considered six components and contaminants are shown by sampling points in Figure 7. The lowest ash content is typical for PET bottles at all points and is confirmed to be 0.2–7.9%. For inert materials such as glass or stones, the ash content on a dry basis is 100%.

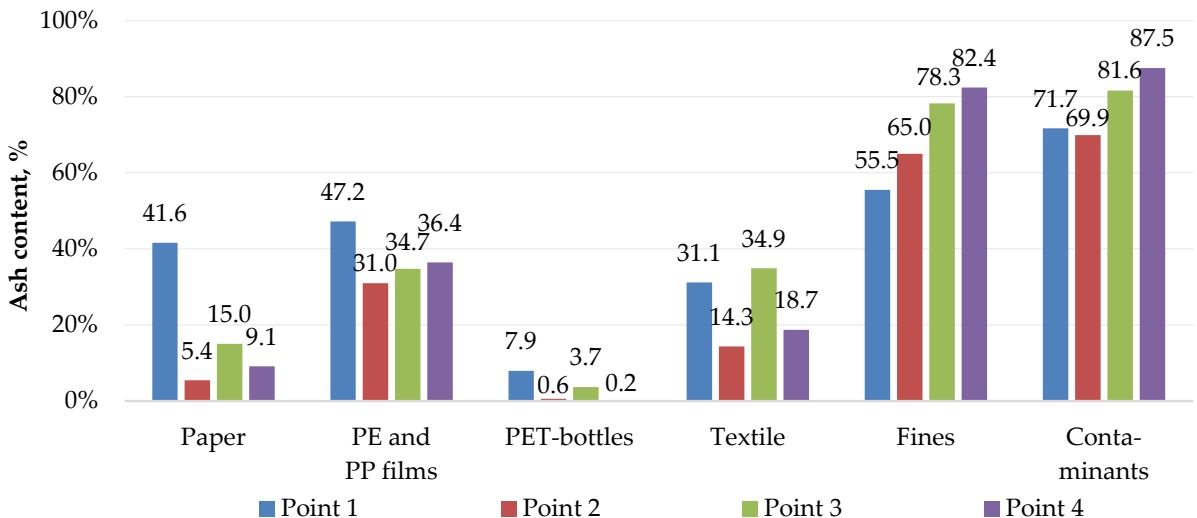

**Figure 7.** Ash content of components on a dry basis.

Figure 6 shows that combustible components, such as PET bottles, PE and PP films, textiles, and waste paper, have rather low ash values. Contaminants have high ash values (69.9–87.5%), as does sifting (55.5–82.4%). This is explained by the fact that contaminants include a combustible component (decomposing organic waste) and an inert component (sand), which adheres to the component when mixed with decaying organic waste.

### 3.5. Component Calorific Value

Based on the obtained results studying moisture content, ash content, and contamination of individual components, the net calorific value of the individual components and excavated waste was calculated for each sampling point. Figure 8 shows the estimated calorific value on a wet basis of individual components with contaminants and in a "clean" state (without contamination) at each sampling point.

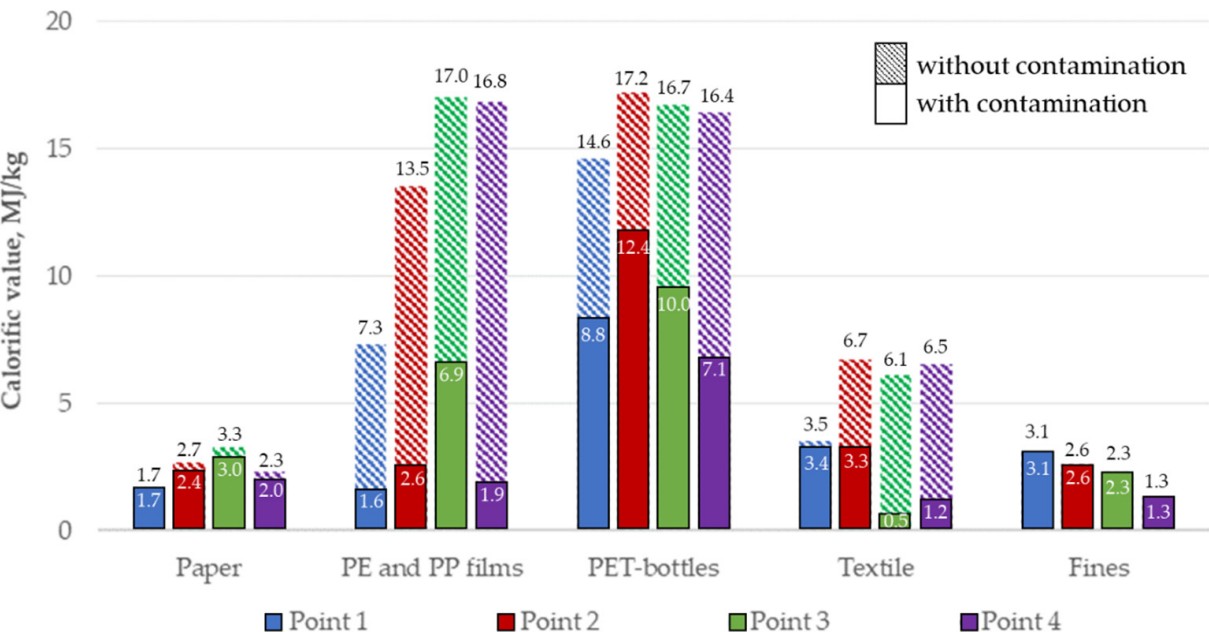

**Figure 8.** Net calorific value on a wet basis of individual components with and without contamination.

Based on the data (Figure 8), it can be concluded that the presence of contaminants in the components significantly affects their calorific value on a wet basis. For example, for PE and PP films at sampling points 3 and 4, the net calorific value on a wet basis differs significantly, even though the moisture content (Figure 5) and ash content on a dry basis (Figure 7) are almost the same. However, the contamination content is 18% and 58% at sampling points 3 and 4 respectively, which explains such discrepancies in the net calorific value on a wet basis.

The highest net calorific value on a wet basis of textiles is 3.4 MJ/kg at sampling point 1 with a maximum moisture content of 67% and a high ash content of 31%, compared to lower values of moisture and ash content on textiles at other points. The presence of contamination on textiles at sampling point 1 is minimal (1.2%), while the contamination content on textiles at points 2, 3, and 4 is 17%, 26% and 27%, respectively.

Figure 8 shows that the net calorific values on a wet basis of individual components at the points are approximately in the same range and depend on the moisture and ash content of each component, which is statistically confirmed by Pearson correlation matrix (Table 3). Net calorific value on a wet basis is related more to the ash content ($r = -0.691$ and $r = -0.720$) than to the moisture content ($r = -0.539$ and $r = -0.535$).

**Table 3.** Pearson correlation matrix.

| Variables | Component Content | Ash Content on a Dry Basis | Moisture Content | Calorific Value on a Wer Basis | Calorific Value on a Wet Basis without Contamination |
|---|---|---|---|---|---|
| Component content | 1 | −0.231 | 0.847 | −0.183 | −0.229 |
| Ash content on a dry basis | −0.231 | 1 | 0.071 | −0.691 | −0.720 |
| Moisture content | 0.847 | 0.071 | 1 | −0.539 | −0.535 |
| Calorific value on a wet basis | −0.183 | −0.691 | −0.539 | 1 | 0.843 |
| Calorific value on a wet basis without contamination | −0.229 | −0.720 | −0.535 | 0.843 | 1 |
| Component content | 1 | −0.231 | 0.847 | −0.183 | −0.229 |

The most combustible components are PET bottles, as well as PE and PP films. The net calorific value on a wet basis of these components ranges from 14.6 to 17.2 MJ/kg and from 7.3 to 17.0 MJ/kg, respectively. The net calorific value on a dry ashless basis of PE and PP films is much higher than the net calorific value of PET bottles, but due to the high moisture content (39–64%), the net calorific value of the films (on a wet basis) is lower than the net calorific value on a wet basis of the PET bottles. Glass has a net calorific value of zero due to the structure of the material. However, the combustible paper itself contained in the excavated landfill materials has a rather low net calorific value on a wet basis, since the moisture content exceeds 66%. The low net calorific value on a wet basis of the sifting (1.3–2.6 MJ/kg) is due to the high ash content (up to 82%).

Results showing the net calorific value on a wet basis of excavated waste calculated as a whole at each point are presented in Table 4.

**Table 4.** Net calorific value on wet basis of excavated waste.

| Sampling Point | Net Calorific Value on a Wet Basis (MJ/kg) |
|---|---|
| Point 1 | 5.03 |
| Point 2 | 4.43 |
| Point 3 | 5.52 |
| Point 4 | 4.26 |
| Average | 4.87 |

The highest net calorific value on a wet basis of the excavated waste is equal to 5.5 MJ/kg at sampling point 3, even though the proportion of combustible components in these samples is slightly higher than other samples. This is due to the fact that samples of this point have a greater content of high-calorie combustible components (polyethylene, etc.). Using the data on component composition, moisture content, ash content, and the presence of contaminants, the average net calorific value on a wet basis of the excavated waste for the facility in question was calculated to be 4.9 MJ/kg, i.e., approximately 1.8 times lower than the net calorific value on a wet basis of untreated MSW (9.0 MJ/kg) [34].

In accordance with GOST R 55127-2012 (CEN/TR 15508:2006) "Solid fuel from household waste. The main properties for compiling a classification system" [35] coal has a calorific value of 17 MJ/kg on average, while solid fuel from MSW has an average of 11.7–25.5 MJ/kg when ash content is low and 3.2–10 MJ/kg when ash content is high. Comparing values presented in Figure 8 with indicators of the calorific value of the fuel (coal), it can be concluded that excavated landfill material cannot be used as secondary fuel without preliminary treatment.

The properties of excavated landfill waste were compared with thermal properties of "fresh" MSW of the city of Perm (Figure 9).

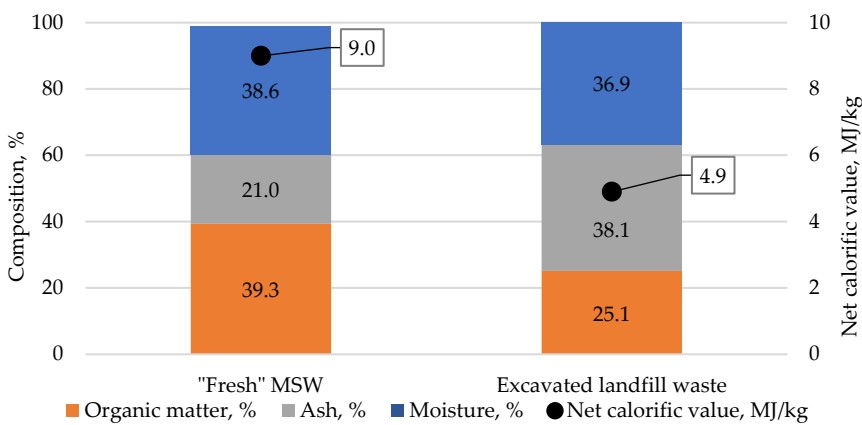

**Figure 9.** Thermal properties of "fresh" MSW and excavated landfill waste.

Net calorific value on a wet basis of excavated landfill waste is 1.8 times lower than a net calorific value on a wet basis of "fresh" MSW. This is due to the fact that the organic content of the excavated landfill waste converted to emissions (leachate and landfill gas). Accordingly, the share of the mineral part of excavated waste has increased and amounts to 38.1%.

Excavated landfill material contains 2.8% metals, which correlates with previously reviewed studies showing 0.4–3.5% [8,21,22,31,32]. Metals can be extracted using separators before being sent for recycling. The content of glass and inert materials in excavated landfill waste is 12% and 7%, respectively, Inert materials can be used as intermediate layers at waste disposal sites.

Excavated landfill material contains 31–46% caloric components (waste paper, PE and PP films, metallized and multilayer films, PET bottles, 3D plastics, textiles, and wood), which can be directed into a separate stream to obtain an energy fraction with a moisture content of 44%, an ash content on a wet basis of 16.9%, and a net calorific value on a wet basis of 7.5 MJ/kg. When dried to a moisture content of 15%, it is possible to obtain fuel with a calorific value of 14 MJ/kg.

The use of secondary fuel from an excavated landfill stockpile could partially or completely replace traditional fuel. The disadvantage of this alternative fuel is the complex and time-consuming technology needed for its manufacture. This includes landfill material excavation, manual or automatic sorting to isolate combustible components, grinding and drying these components, and finally, forming briquettes. To improve the quality of alternative fuel from landfill materials, other waste with a high net calorific value on a wet basis can be added to its composition.

## 4. Conclusions

For the first time, it was proposed to include a stage of removing contaminants from the surface of excavated waste components when determining the waste composition and thermal properties of excavated waste. This allows researchers to get more accurate data on the properties of waste components and their contaminants. For example, the fraction of adhering contaminants can reach up to 33.4% (on a dry basis) for metallized and combined materials, with moisture content of 45%, which is lower than the moisture content of contaminants (54%). Therefore, the net calorific value of metallized and combined materials is differed significantly with and without contamination. As a result, it is possible to estimate the net calorific value of waste without overestimation both for the mixed excavated waste flow and for the flow of combustible components.

The thermal properties of excavated waste were analyzed for the first time in the Russian Federation. But the situation is similar for many other countries. Waste management system in RF, as in many other countries, are based on mixed waste disposal. Many waste disposal sites have no engineered infrastructure (base liner, intermediate covering, final

cover, leachate and landfill biogas collection systems, etc.). There is no system of separate waste collection, so the landfilled waste has a modern composition with a high proportion of polymers. In this regard, waste can be considered as an energy resource.

Detailed study of the composition and thermal properties of components of landfill material allow a fairly accurate assessment of its resource potential. The content of combustible components in the excavated waste is 31–46.6%.

Among combustible components with the highest moisture content are waste paper (69.1%) and diapers (65.8%), whereas PET bottles (3.1%), wood (11.2%), and other 3D plastics (13.4%) have rather low ash content on a dry basis. These indicators are reflected in the net calorific value of components and ultimately determine the energy potential of excavated waste as a whole.

The identifiable components of the excavated landfill material are significantly contaminated (the mass of contamination is up to 36% at moisture content 54.3% and ash content on a dry basis 78.5%), which must be taken into account when determining the composition of landfill material and its properties. The net calorific value of the individual components and excavated waste as a whole depends not only on the moisture and ash content of the individual components, but also on the presence of contaminants in these components.

The average net calorific value on a wet basis of the excavated waste is 4.9 MJ/kg, and for the separate mixture of combustible components, it is 7.5 MJ/kg at a moisture content of 44%. Net calorific value on a wet basis of "fresh" MSW is 9.0 MJ/kg, which is comparable to the prepared flow of combustible components of excavated waste. This means that excavated landfill materials can be regarded as a resource for the manufacture of secondary fuel only after pretreatment that includes at least sorting and drying. It is important to mention that with the help of mechanical sorting, it is possible to remove impurities from components; it could increase the calorific value of waste combustion.

**Author Contributions:** Conceptualization, S.P. and G.I.; methodology, G.I., N.S. (Natalia Sliusar) and V.K.; investigation, I.S. and S.P.; data curation, V.K. and N.S. (Nemanja Stanisavljevic); writing—original draft preparation, I.S., S.P. and I.M.; writing—review and editing, N.S. (Natalia Sliusar), G.I., V.K. and N.S. (Nemanja Stanisavljevic); visualization, S.P. and G.I.; supervision, N.S. (Natalia Sliusar) and N.S. (Nemanja Stanisavljevic); project administration, G.I.; funding acquisition, N.S. (Natalia Sliusar). All authors have read and agreed to the published version of the manuscript.

**Funding:** This research was funded by Ministry of science and higher education of the Russian Federation (Project № FSNM-2020-002).

**Institutional Review Board Statement:** Not applicable.

**Informed Consent Statement:** Not applicable.

**Data Availability Statement:** The study did not report any data.

**Conflicts of Interest:** The authors declare no conflict of interest. The funders had no role in the design of the study; in the collection, analyses, or interpretation of data; in the writing of the manuscript, or in the decision to publish the results.

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
