# Peer review of "Energy Potential Assessment of Excavated Landfill Material: A Case Study of the Perm Region, Russia"

_recycling, doi:10.3390/recycling7010007_

Round 1

Reviewer 1 Report

The article touches undoubtedly a critical issue, especially in the view of hazards related to unprotected landfills and the potential value of buried waste as stocks of materials. Therefore, the subject addressed in this article is definitely worthy of investigation.

In general, the article is well-written and concise. Providing an additional (Russian) perspective on landfill mining and the discussion of excavated waste potential are additional strengths of the manuscript. However, some clarifications are necessary before the publication. Please see below some points I recommend to consider for further revision:

  1. You refer to studies in your manuscript without mentioning the authors' names (e.g., lines 63, 73,95). I recommend using 'in a study by Särkkä et al. [19] / Bhatnagar et al. study [21] / Studies of Silusar et al. [7,26]'.
  2. How did you ensure the representativeness of the waste samples (1 kg) taken for further studies? Did you follow the procedure of any standard, such as, for instance, EN 15413:2011?
  3. The soil/inert material/contaminants are often pressed into soft waste surfaces (e.g., plastics). As you performed brushing of the waste, you removed part of the contamination. Is it then accurate to call it a 'level of contamination? Isn't it more a measure of detachable dirt/contaminants amount? This value might be beneficial as the detachable dirt/contaminants might be partially removed during pre-processing of waste (e.g., mechanical sorting), thus increasing waste's calorific and commercial value.
  4. The sentences in lines 145-148 are not clear.
  5. What does unspent means (line 154)?
  6. How did you measure 'the lowest calorific value for dry unspent mass of the ? component' (line 154)? Why did you take into account the lowest value, not the average value measured?
  7. The sentence in lines 176-179 needs clarification – why is it not 'high enough'?
  8. Line 185 – Table 2 (the upper case).
  9. In lines 260-263, you state that you determine the ash and moisture contents for the contaminants. Where are the results of it?
  10. In lines 345-353, you discuss the possible use of the mentioned fractions, and prior to this, you discuss metals. What about the other fractions, like, for instance, inert material or glass?
  11. Line 354 – 'The use of secondary fuel from an excavated landfill stockpile could/have potential to partially or completely replace traditional fuel'. 'Will' seems to be a bit of an overstatement.

Author Response

Dear Reviewer, 

thank you for your valuable comments. Please, find our response in the attached file

Reviewer 2 Report

The manuscript presents a study case of landfill waste excavation in Russia with focus on the fuel properties of the excavated waste. The manuscript presents interesting data on the Russian waste composition and properties which can be useful in promoting the landfill mining framework. A clear introduction has been written to highlight the differences between Russian landfill and other European landfills. Nevertheless, the manuscript can be improved further.

The following are more details comments from me which might be useful in improving the manuscript.

  1. I suggest the authors to highlight or adding the location of the landfill mining to the paper title and abstract. 
  2. In the method section, it is said that the samples were taken from 4 different locations of the landfill. Are there any specific reasons of choosing these 4 locations? Please explain it in the manuscript.
  3. A further discussion is needed regarding the comparison of the current study with the literature having similar methods. This is especially important to highlight the unique findings of the study. The authors have compiled a summary on the landfill waste composition from different literature in Table 2. However detail discussion is missing (only a different in fine fraction has been presented). The following aspects can be added in the discussion:
    • Is there any differences in the major material compositions compared to the other landfills?
    • How is the potential hazardous compounds found in the landfill?
    • Is there any differences in the fuel properties of waste compared to the other landfills?
    • Are there any specific reasons considering the locations, landfill policy, and surrounding populations that affects the differences above?
  4. The authors presents overall material compositions and fuel properties based on the waste from different landfill depths. Can they discuss the effect of the depth on the landfill waste properties?
  5. The conclusion should also contains the summary of the discussion on point 3 above.

Author Response

(The authors gave the same response as above.)

Reviewer 3 Report

I apologize for the comments that I am going to make. Personally, I consider that the work carried out corresponds to a Technical Report, rather than an investigation.

As a Technical Report, I think there are areas for improvement:

The references of the moisture percentages is confusing when it refers to the ash fractions.

The calorific values ​​must refer to whether they are based on dryness or humidity in all cases.

The chemical composition of the possible ash that could be obtained should be determined, as well as experimentally determine the calorific values  of some fractions to be able to compare with the estimated ones.

The results  should have been compared experimentally with fresh residues.

Author Response

(The authors gave the same response as above.)

Round 2

Reviewer 1 Report

Thank you for addressing my comments. I have no further comments to add.

Reviewer 2 Report

The authors have made some changes on the manuscript. However, I do not think the authors have fully response the reviewers' comments with sufficient answers and revisions. I suggest the authors to review again previous comments and carefully address them.

Here are my comments on their response.

  1. In my previous comments, I have emphasized that the authors should highlight the unique or new findings related to waste obtained from their landfill site compare to the previous studies involving different landfill sites. This should be done throughout their whole manuscript including the discussion, and more importantly the conclusion. Frankly speaking, there is not any novelty related to their method and approach. The presented data related to the measured waste properties also do not have any new findings, it is just another data from another landfill site. So, please shows what is your main conclusion regarding your excavated waste in respect to the geographical location, existing landfill management etc.
  2. Please add references to back up your arguments (as shown in the next sentence) and please add this explanation to the manuscript, and what are the examples on these southern countries. "Response 3: In southern countries, the share of fine fraction is higher compared to European countries. Since in European countries the share of packaging waste (plastic, glass, paper, metal) prevails significantly even taking into account the ban for disposal valuable components in landfills."
  3. You did not answer my previous comment as follow: The authors presents overall material compositions and fuel properties based on the waste from different landfill depths. Can they discuss the effect of the depth on the landfill waste properties?

Author Response

We appreciate the comments from the reviewer, which help us to prepare a 
better manuscript. 

Reviewer 3 Report

The paper  is acceptable for publicaction

Author Response

(The authors gave the same response as above.)

Round 3

Reviewer 2 Report

The authors have provided satisfactory answers to the reviewers' comments. I appreciate their effort in significantly improving their manuscript. I suggest to accept their manuscript for a publication.